# Enhancing 6-DoF Object Pose Estimation through Multiple Modality Fusion: A Hybrid CNN Architecture with Cross-Layer and Cross-Modal Integration

Zihang Wang [1], Xueying Sun [1,2,*], Hao Wei [3], Qing Ma [1] and Qiang Zhang [1,2]

1 College of Automation, Jiangsu University of Science and Technology, No. 666 Changhui Road, Zhenjiang 212100, China; 202210305124@stu.just.edu.cn (Z.W.); 202210305204@stu.just.edu.cn (Q.M.); qzhang@just.edu.cn (Q.Z.)
2 Systems Science Laboratory, Jiangsu University of Science and Technology, No. 666 Changhui Road, Zhenjiang 212100, China
3 Shenlan College, Jiangsu University of Science and Technology, No. 666 Changhui Road, Zhenjiang 212100, China; 192211205138@stu.just.edu.cn
* Correspondence: sunxueying@just.edu.cn

**Abstract:** Recently, applying the utilization of RGB-D data for robot perception tasks has garnered significant attention in domains like robotics and autonomous driving. However, a prominent challenge in this field lies in the substantial impact of feature robustness on both segmentation and pose estimation tasks. To tackle this challenge, we proposed a pioneering two-stage hybrid Convolutional Neural Network (CNN) architecture, which connects segmentation and pose estimation in tandem. Specifically, we developed Cross-Modal (CM) and Cross-Layer (CL) modules to exploit the complementary information from RGB and depth modalities, as well as the hierarchical features from diverse layers of the network. The CM and CL integration strategy significantly enhanced the segmentation accuracy by effectively capturing spatial and contextual information. Furthermore, we introduced the Convolutional Block Attention Module (CBAM), which dynamically recalibrated the feature maps, enabling the network to focus on informative regions and channels, thereby enhancing the overall performance of the pose estimation task. We conducted extensive experiments on benchmark datasets to evaluate the proposed method and achieved exceptional target pose estimation results, with an average accuracy of 94.5% using the ADD-S AUC metric and 97.6% of ADD-S smaller than 2 cm. These results demonstrate the superior performance of our proposed method.

**Keywords:** cross layer; cross modality; hybrid CNN architecture; object pose estimation

## 1. Introduction

Accurately estimating the six-degree-of-freedom (6-DoF) of objects is a critical task in various applications, including robotics, autonomous driving, and virtual reality. For instance, the precise estimation of spatial coordinates and rotational orientation of an object is essential for robotic tasks such as manipulation, navigation, and assembly. However, achieving robustness in 6-DoF detection remains a challenging problem. In real-world applications, numerous object types exhibit significant occlusions and variations in lighting conditions. Due to the increasing reliability of new RGB-D image sensors, the 6-DoF detection of visual targets based on multi-source image information is flourishing. Researchers have explored a number of ways [1–3] to fuse RGB image data and depth image data to guide 6-DoF detection of visual targets with impressive accuracy. Different research teams are employing various framework approaches to investigate solutions for the 6DoF pose estimation problem. Some focus on the overall algorithmic framework, while others delve into efficient feature extraction. In contrast, our emphasis lies in the efficient extraction of features from the target object. To enhance model interpretability in the framework selection, we opted for the classical two-stage framework.

Regarding the problem of object pose estimation, previous approaches predominantly employed adaptive matrices to tackle this issue. However, with the rise of convolutional neural networks (CNN) and transformers, deep learning (DL) based methods are used to solve the 6-DoF estimation problem. There are two main types of DL-based frameworks for 6D attitude estimation of objects: end-to-end architectures [4,5] and two-stage segmentation-pose regression architectures [6,7]. End-to-end models integrate multiple stages of visual processing steps into a single model; therefore, their networks are less complex and computationally intensive. A single network processes pixel information from the image to deduce the region where the candidate target is located and its corresponding 6DoF pose information. The internal structure and decision-making process of this neural network are more hidden, less interpretable, and more difficult to train. On the other hand, the two-stage segmentation-pose regression architecture first segments the visual target from the scene and then obtains the pose of the visual target in the scene by regression. This method is able to focus on the visual target being detected and exclude interference from the background, resulting in more reliable results.

In the process of 6DoF pose estimation through image features, there have been numerous prior efforts. Some have employed manually designed features (such as SIFT) to extract object characteristics for subsequent pose regression. However, the limited quantity of manually designed features might lead to failures in pose regression. Depth images provide dense features, yet enhancing the robustness of these depth features remains an unsolved challenge. Solely relying on RGB or depth information addresses only one facet of the problem. Thus, the approach in this study leverages the fusion of RGB-D data to accomplish the task. Prior research has made significant strides in exploring the fusion of RGB and depth images. A multitude of studies have delved deeply into various techniques and algorithms aiming to effectively exploit the complementary information these modalities provide. However, despite these commendable efforts, achieving seamless integration between RGB and depth images remains an ongoing and formidable challenge. Existing methods often grapple with the intricate task of synchronizing the two modalities accurately, resulting in less than optimal fusion outcomes. Moreover, inherent differences in intrinsic features between RGB and depth data, including variations in lighting conditions and occlusions, further amplify the complexity of the fusion process. As such, continuous research and innovation are urgently needed to elevate the fusion of RGB and depth images in target pose detection to new heights.

In order to enhance the precision of 6-DoF pose detection, we presented a novel integrated two-stage framework. This framework combined a semantic segmentation network with a 6-DoF pose regression network, thereby facilitating the comprehensive integration of pertinent information derived from both RGB and depth images. By leveraging this approach, we aimed to extract more efficient and robust deep learning features, ultimately leading to improved accuracy in 6D object pose estimation. The integration of the semantic segmentation network and the regression network enabled a synergistic fusion of the complementary strengths of both modalities, resulting in enhanced reliability and performance in 6-DoF detection. The contributions of our work are listed below:

First, to enhance the accuracy of segmentation, a novel Three-Flow fusion network was proposed. This architecture incorporates Cross Layer Spatial-wise attention and Cross Modality Spatial-wise attention CNN mechanisms to address the challenge of multimodal data fusion and guide the segmentation of candidate targets. Meanwhile, in order to verify the validity of the proposed mechanisms, we conducted detailed ablation experiments for each of the proposed mechanisms based on the idea of post hoc interpretability [8,9] and verified the validity of these mechanisms, as detailed in Section 4.4.2.

Second, a CBAM-based feature enhancement network was introduced to improve the robustness of 6-DoF pose regression for visual targets.

Finally, the proposed method improved upon a leading method by achieving greater scores in the most used YCB-Video dataset in the 6D pose estimation area. Our method

achieved on average 94.5% using ADD-S AUC metric and 97.6% of ADD-S smaller than 2 cm.

The article is structured as follows: Section 2 offers a comprehensive review of prior research on object pose estimation. Section 3 outlines the intricacies of our proposed approach. In Section 4, we present a thorough performance evaluation of our method. Lastly, Section 5 provides a comprehensive summary and conclusion of the article.

## 2. Related Works

### 2.1. Feature Representation

In vision tasks, the representation of image features plays a crucial role in various applications, including visual target recognition and detection. In the context of target pose estimation, it is essential for the features of visual targets to exhibit robustness against translation, rotation, and scaling. Additionally, these features should possess local descriptive capabilities and resistance to noise.

In previous studies, researchers have utilized image feature matching to detect the position of visual targets. The pose of the target can be obtained by solving the 2D-to-3D PnP problem. Artificially designed features such as SIFT [10,11], SURF [12], DAISY [13], ORB [14], BRIEF [15], BRISK [16], and FREAK [17] have demonstrated robustness against occlusion and scale-scaling issues. These descriptors have been widely adopted in models for target position detection. Similarly, 3D local features such as PFH [18–20], FPFH [21], SHOT [22], C-SHOT [23], and RSD [24] can effectively extract features and detect the position of targets in 3D point clouds. Recently, machine learning based feature descriptor algorithms [25,26] are receiving more and more attention in the field of image matching. These methods employ PCA [27], random trees [28], random fern [29], and boosting [30] algorithms to achieve more robust features than hand-designed features.

However, in cases where the surface of the visual target is smooth and lacks texture, the extraction of manually designed feature points is often limited in number. This limitation adversely affects the reliability of object pose estimation. Furthermore, the high apparent similarity among visual targets also poses challenges in accurately estimating the positional attitude of the detected target.

In addition to manually designed features, there are supervised learning-based feature description methods such as triplet CNN descriptor [31], LIFT [32], L2-net [33], Hard-Net [34], GeoDesc [35]. For the recognition of textureless objects, global features can be implemented by utilizing image gradients or surface normals as shape attributes. Among these, template-based global features aim to identify the region in the observed image that bears the closest resemblance to the object template. Some commonly employed template-based algorithms include Line-MOD [36] and DTT-OPT [37]. In recent years, novel 3D deep learning methods have emerged, such as OctNet [38], PointNet [39], PointNet++ [40], and MeshNet [41]. These methods are capable of extracting distinctive deep representations through learning and can be employed for 3D object recognition or retrieval.

### 2.2. Two-Stage or Single-Shot Approach

In the realm of object 6D pose estimation frameworks, two main types can be identified: end-to-end architectures and two-stage segmentation 6-DoF regression architectures.

In the field of object detection, notable end-to-end frameworks like YOLO [42] and SSD [43] have emerged. These frameworks have been extended to address the challenge of target pose detection. Poirson et al. [44] proposed an end-to-end object and pose detection architecture based on SSD, treating pose estimation as a classification problem using RGB images. Another extension, SSD-6D [45], utilizes multi-scale features to regress bounding boxes and classify pose into discrete viewpoints. Yisheng He et al. [46] introduced PVN3D, a method based on a deep 3D Hough voting network that fuses appearance and geometric information from RGB-D images.

Two-stage architectures segment the visual target and estimate pose through regression. For example, in [47], pose estimation was treated as a classification problem using the

2D bounding box. Mousavian et al. [48] utilized a VGG backbone to classify pose based on the 2D bounding box and regress the offset. Nuno Pereira et al. [7] proposed Masked-Fusion, a two-stage network that employed an encoder–decoder architecture for image segmentation and utilized fusion with RGB-D data for pose estimation and refinement. This two-stage neural network effectively leverages the rich semantic information provided by RGB images and exhibits good decoupling, allowing for convenient code replacement when improvements are required for a specific stage algorithm. Additionally, this design helps reduce training costs.

However, the MaskedFusion method employed in the first stage solely relies on RGB image information, which often leads to insufficient and inaccurate semantic information in low-light and low-texture scenarios. This results in issues such as blurry edges and erroneous segmentation in the Mask image of the segmentation network during practical scene applications.

### 2.3. Single Modality or Multi-Modality Fusion

### 2.3.1. RGB Single Modal Based Object Pose Estimation

For visual target position detection, RGB images have traditionally been used as the primary data source. Feature matching techniques are commonly employed for localizing target positions within 2D images. PoseCNN [49] utilizes a convolutional neural network and Hough voting to estimate the target's pose. PvNet [50] extracts keypoints from RGB images and employs a vector field representation for localization. Hu et al. [51] proposed a segmentation-driven framework that uses a CNN to extract features from RGB image and assigns target category labels to virtual meshes. The ROPE framework [52] incorporates holistic pose representation learning and dynamic amplification for accurate and efficient pose estimation. SilhoNet [53] also predicts object poses using a pipeline with a convolutional neural network. Zhang et al. [54] proposed an end-to-end deep learning architecture for object detection and pose recovery from single RGB modal data. Aing et al. [55] introduced informative features and techniques for segmentation and pose estimation.

Although image-based methods have achieved promising results in 6-DoF estimation, their performance tends to degrade when dealing with textureless and occluded scenarios.

### 2.3.2. 3D Cloud or Depth Image Based Object Pose Estimation

Recovering the position of a visual target from 3D point cloud or depth image data is also a common method. The RGM method [56] introduces deep graph matching for point cloud registration, leveraging correspondences and graph structure to address outliers. This approach replaces explicit feature matching and RANSAC with an attention mechanism, enabling an end-to-end framework for direct prediction of correspondence sets. Rigid transformations can be estimated directly from the predicted correspondences without additional post-processing. The BUFFER method [57] enhances computational efficiency by predicting key points and improves feature representation by estimating their orientation. It utilizes a patch-wise embedder with a lightweight local feature learner for efficient and versatile piecewise features. The ICG framework [58] presents a probabilistic tracker that incorporates region and depth information, relying solely on object geometry

Nonetheless, the point cloud data inherently exhibits sparsity and lacks sufficient texture information, which poses limitations to the performance of these methods. Consequently, the incorporation of RGB image information represents a crucial enhancement to enhance the accuracy and effectiveness of the position estimation.

### 2.3.3. Multi-Modal Data Based Object Pose Estimation

In the field of target position detection, the fusion of information from multiple sensors has emerged as a cutting-edge research area for accurate position detection. Zhang et al. [59] proposed a hybrid Transformer-CNN method for 2-DoF object pose detection. They further proposed a bilateral neural network architecture [60] for RGB and depth image fusion and

achieved promising results. In 6-DoF pose detection area, Wang et al. [6] introduced the DenseFusion framework for precise 6-DoF pose estimation using two data sources and a dense fusion network. MaskedFusion [7] achieved superior performance by incorporating object masking in a pipeline. Se(3)-TrackNet [61] presented a data-driven optimization approach for long-term 6D pose tracking. PVN3D [46] adopted a keypoint-based approach for robust 6DoF object pose estimation from a single RGBD image. FFB6D [5] introduced a bi-directional fusion network for 6D bit-pose estimation, exploiting the complementary nature of RGB and depth images. The ICG+ [62] algorithm incorporated additional texture patterns for flexible multi-camera information fusion. However, existing methods still face challenges in extracting feature information from RGB-D data.

## 3. Method

### 3.1. Network Overview

As mentioned previously, the accurate detection of the six degrees of freedom pose for visual targets relies on effectively utilizing semantic information to separate the targets from the background and obtain valuable and valid pixel regions. This is crucial for guiding subsequent target pose prediction. Furthermore, leveraging the information from RGB and depth images to create robust feature representations is essential for accurate target pose inference.

To address these challenges, we proposed the two-stage object pose detection framework, which combined semantic segmentation and pose estimation. As depicted in Figure 1, the proposed architecture was composed by the semantic segmentation network and object pose prediction network. Our work enhanced the segmentation process by integrating CM module for RGB and depth information interaction and the CL module for inter-feature layer guidance. These modules improved the segmentation approach during the segmentation stage. Additionally, in the subsequent target pose regression stage, we introduced the CBAM attention mechanism, which densely embedded attention into features, enhancing their robustness and reducing the influence of interfering features. This increased the reliability of pose regression. The visual object pose detection framework proposed in this study is illustrated in Figure 1.

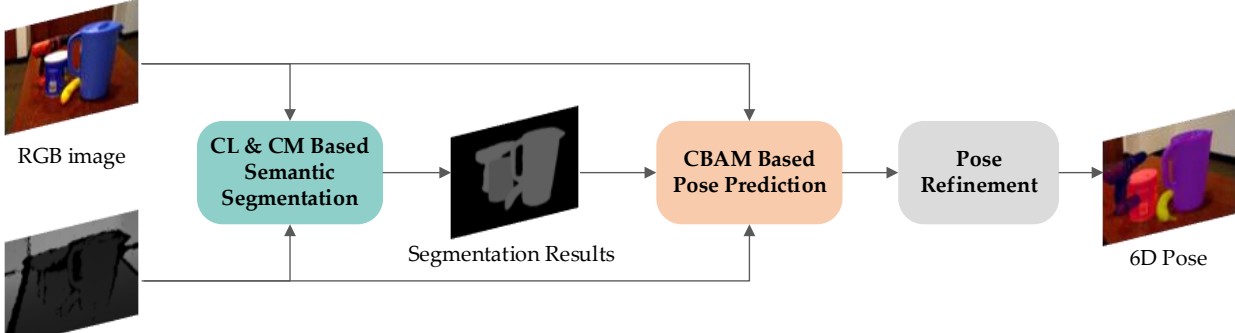

**Figure 1.** Proposed network architecture in this study. The proposed method has three sub-tasks: CL and CM-based semantic segmentation, CBAM-based prose prediction and pose refinement.

As shown in Figure 1, the framework first input RGB-D data into our proposed segmentation network to extract features (details in Section 3.2). These features were then passed through a dense feature embedding module with the CBAM attention mechanism (details in Section 3.3.1). A similar object pose prediction approach to DenseFusion was employed for pose estimation and tuning (details in Section 3.3.2).

Specifically, in the context of RGB-D data, we employed Convolutional Neural Network (CNN) architectures to extract features independently from the RGB map and the depth map, resulting in two distinct feature streams. These streams were mutually guided by the Cross-Modal (CM) module during the feature extraction process, facilitating effective information exchange between the RGB data and the depth data. Additionally, a fusion layer was introduced to integrate the RGB streams and depth streams, generating a third stream that served as an intermediary between the RGB streams and the feature streams. Each of these three branches, comprising the aforementioned feature streams, can be regarded as a feature extraction network with a structure akin to U-Net [63] (which is a classical encoder–decoder structure, but innovatively includes mechanisms for encoding features to guide decoding features). Finally, for the regression and refinement of 3D rotations and 3D translations of objects, we employed a CNN network equipped with adaptive point cloud quantity sampling and a spatial and channel-based attention mechanism.

### 3.2. CM and CL Module-Based Three-Stream Segmentation Network

In the realm of 6DoF pose estimation, previous research has largely overlooked the potential loss of information that occurs during the feature extraction stages. It is widely recognized that low-level feature maps encompass valuable image details, and it is imperative to devise an effective approach that preserves spatial information while retaining the original image information in higher-level feature maps. To address this critical concern, we proposed a novel cross-layer (CL) strategy in this study. The primary objective of this strategy was to mitigate the loss of information during feature extraction by seamlessly integrating image details from low-level feature maps into their higher-level counterparts. By incorporating this CL strategy, we aimed to enhance the overall accuracy and robustness of 6DoF pose estimation by preserving crucial image details throughout the feature extraction process.

Moreover, the prevailing studies in object pose detection have predominantly relied on RGB data for segmentation purposes. However, it is important to acknowledge that RGB images alone may exhibit limitations in scenarios characterized by weak texture, poor lighting conditions, or objects with transparency. In contrast, depth images possess the potential to compensate for these limitations by providing additional information that complements the RGB features. Hence, it becomes imperative to explore methodologies that facilitate meaningful information exchange between calibrated RGB data and depth information. To address this challenge, we proposed the integration of a cross-modal (CM) module in our research. This CM module served as a pivotal component in our segmentation network framework, as depicted in Figure 2.

The segmentation network proposed in this study leverages RGB-D data as input, enabling a comprehensive analysis of both modalities. To facilitate this analysis, the network establishes separate feature streams for RGB and depth images. At the initial layer of the encoder, a fusion process is employed to integrate RGB and depth features, resulting in a fused feature stream. Throughout the encoder layers, the three branches of the network engage in multimodal information interaction, facilitated by the Cross-Modal (CM) module. This module enables effective exchange and integration of information between the RGB and depth streams. Furthermore, to address the challenge of information loss during feature extraction, the Cross Layer (CL) module guides each branch across feature layers, ensuring the preservation of crucial details. The decoder for each branch follows a design inspired by U-Net, facilitating encoder–decoder information interaction and achieving upsampling. Finally, the feature maps from the three branches are fused to produce the network's output, providing accurate and comprehensive segmentation results.

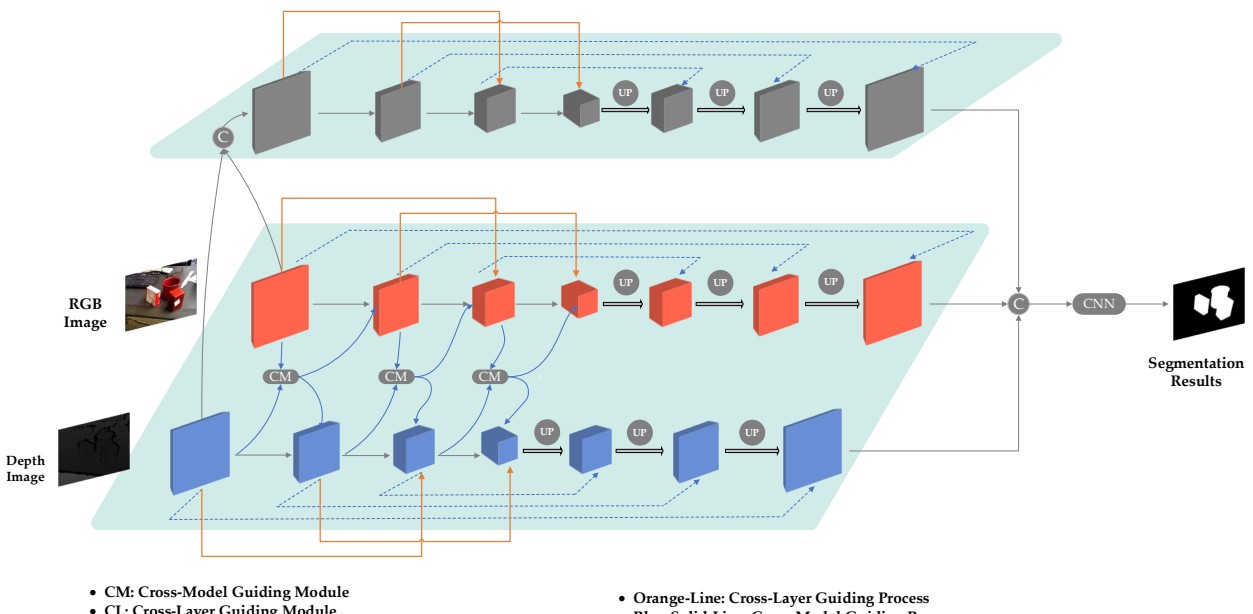

**Figure 2.** Segmentation network framework proposed in this study. A CNN network is utilized for learning of RGB image and depth image, respectively. In flow of three networks, CL and CM = guiding Module are added as bridges for information.

### 3.2.1. Cross Modal Module (CM)

In scenarios characterized by low-texture and low-light conditions, RGB images often exhibit a lack of sufficient feature information, while depth images may lack crucial color and texture details. Consequently, it becomes imperative to establish an effective mechanism for information exchange between the RGB and depth feature extraction streams. To tackle this challenge, we proposed the incorporation of a Cross-Modal Feature Guiding Module (as illustrated in Figure 3) in this section. This module serves as a facilitator for the exchange of information between RGB and depth images, enabling a synergistic fusion of their respective strengths. It is important to note that this research assumes the prior calibration of RGB and depth images, which can be mathematically represented as follows:

$$F_{RSA} = \delta(MLP(AvgPool(F_R)) \oplus (MLP(MaxPool(F_R)))) \tag{1}$$

$$F_{DSA} = \delta(MLP(AvgPool(F_D)) \oplus (MLP(MaxPool(F_D)))) \tag{2}$$

$$F'_R = \delta(F_R \otimes F_{DSA}) \tag{3}$$

$$F'_D = \delta(F_D \otimes F_{RSA}) \tag{4}$$

In this module, the input features for the RGB and depth branches are represented by $F_R$ and $F_D$, respectively. *MaxPool* and *AvgPool* refer to the max-pooling and average-pooling operations, respectively. $\oplus$ stands for channel-wise concatenation. The *MLP* denotes the perception layer, which comprises two convolutional layers followed by the ReLU activation function represented by $\delta$. It is noteworthy that the weights of the RGB and depth branches are not shared during this process. To achieve this, the module initially employs the Spatial Attention (SA) module to extract features from the input feature maps $F_{RSA}$ and $F_{DSA}$. In SA module, $\otimes$ represents the pixel-wise multiplication.

In the segmentation network, the CM module plays a crucial role at each layer of the encoder by facilitating cross-modal information interaction between RGB and depth data. Given that the RGB and depth data are calibrated upon entering the network, the

CM module employs a multiplication operation to fuse the RGB/depth spatial attention map with the depth/RGB feature. This design enables the RGB branch to enhance its feature extraction capability, particularly in environments characterized by weak texture and low lighting conditions. Simultaneously, the depth branch overcomes the limitation of lacking color texture information, thereby achieving a more comprehensive representation. However, it is important to acknowledge that relying solely on the CM module may not accurately extract features for each branch, especially in complex and diverse scenarios. To address this inherent challenge, we developed a unique CM module that effectively handles these complexities, ensuring more accurate and robust feature extraction for both branches.

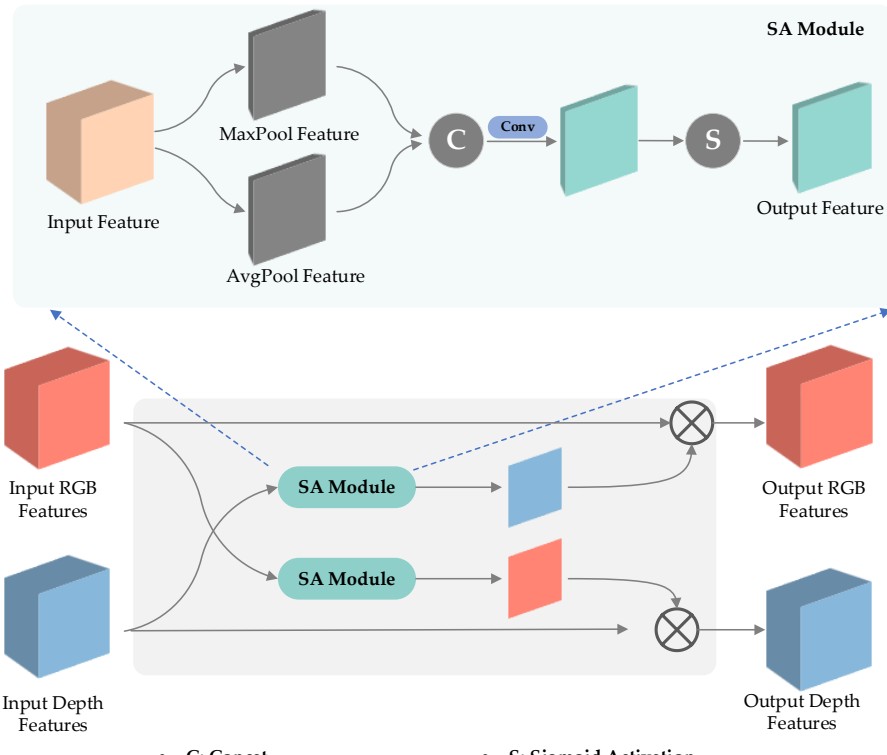

**Figure 3.** Structure of the cross-modal guiding module. For each point, we extracted the spatial attention weights using a fusion of MaxPool and AvgPool, and then used multiplication to fuse point features on the cross-modal feature maps.

### 3.2.2. Cross Layer Strategy (CL)

In this section, we introduce a novel Cross-Layer Guiding Feature Extraction Module (CL) aimed at enhancing the feature extraction capability of the encoder. As shown in Figure 4, we acknowledge that low-level feature maps, characterized by fewer channels but larger size, inherently possess more comprehensive semantic information compared to high-level feature maps, which exhibit more channels but smaller size. Recognizing this distinction, our approach leverages the CL module to facilitate inter-feature layer guidance. By incorporating the CL module, we enabled the exchange of valuable information between different layers, allowing the encoder to effectively leverage the rich semantic details present in the low-level feature maps. This inter-layer guidance mechanism enhances the overall feature extraction process, leading to improved performance and accuracy in subsequent stages of the network.

In this approach, the initial processing of low-level features involves passing them through the spatial attention module, resulting in the generation of a spatial attention graph. To ensure compatibility with high-level features, this attention map undergoes downsampling using a maximum pooling technique, while still preserving the fundamental

semantic information of the low-level features. The downscaled spatial attention map is subsequently multiplied with the high-level feature map, giving rise to a cross-layer bootstrap feature map. This iterative process, as described by Equations (5) and (6), enables the seamless integration of low-level and high-level features, thereby enhancing the overall representation of the input data.

$$F_L = \delta(MLP(AvgPool(F_R)) \oplus (MLP(MaxPool(F_R)))) \tag{5}$$

$$F_H = F_H \otimes \delta(MaxPool(F_L)) \tag{6}$$

In the given formula, $F_L$ denotes the low-level spatial attention features that are extracted using the Spatial Attention Module. On the other hand, $F_H$ represents the high-level features that are guided by the low-level features. By integrating the semantic information derived from the low-level features into the high-level ones, we can effectively mitigate the issue of low efficiency in fitting the overall features within the high-level features. Consequently, this approach accelerates the convergence of network training.

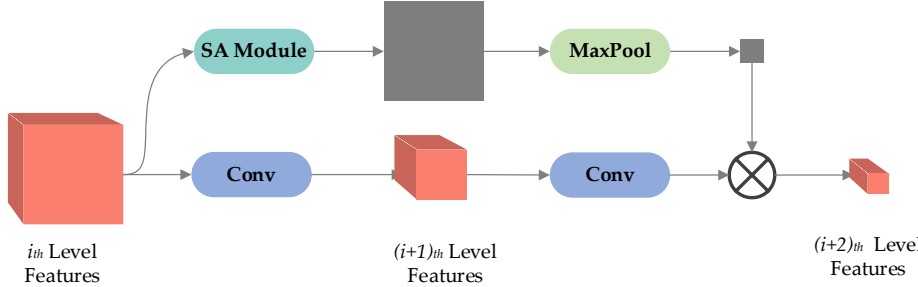

**Figure 4.** Structure of the cross-layer guiding module. The points of the low-level feature map are inputted into the Spatial Attention module to acquire the weight associated with each point in the feature map. Subsequently, we resize it to match the dimensions of the high-level feature map using MaxPool. Finally, point-to-point feature fusion is performed through element-wise multiplication.

In the depicted upsampling process as illustrated in Figure 2, our work incorporated the skip connection design inspired by U-Net. This design choice enables the preservation of a greater amount of contextual information from the encoder layers during the decoding process. The utilization of skip connections is a well-established technique in the field of image processing and is widely recognized for its ability to enhance the performance of the decoding phase. For a visual representation of this design, please refer to Figure 5.

Specifically, the upsampling module serves the purpose of decoding both the low-level image details and high-level semantic features. In order to seamlessly incorporate the semantic features obtained from the encoder stage into the upsampling process of the decoder, a reverse max-pooling method was employed. This method enables the generation of decoder feature maps with identical dimensions and channel numbers as the feature maps in the encoder stage. Subsequently, the two feature maps, having the same dimensions, are combined through element-wise addition. This fusion process effectively integrates the feature maps from both the encoder and decoder stages, resulting in a generated feature map that encompasses both the high-level semantic information from the encoder stage and the low-level image details from the decoder stage. The described process can be summarized by Equation (7).

$$F_{UP} = \delta(conv(F_{DN} \oplus UnMaxpool(F_{EN}))) \tag{7}$$

In Equation (7), $FEN$ represents the feature map from the encoder stage, $FDE$ represents the feature map from the decoder stage, and $FUP$ represents the feature map generated by the upsampling module.

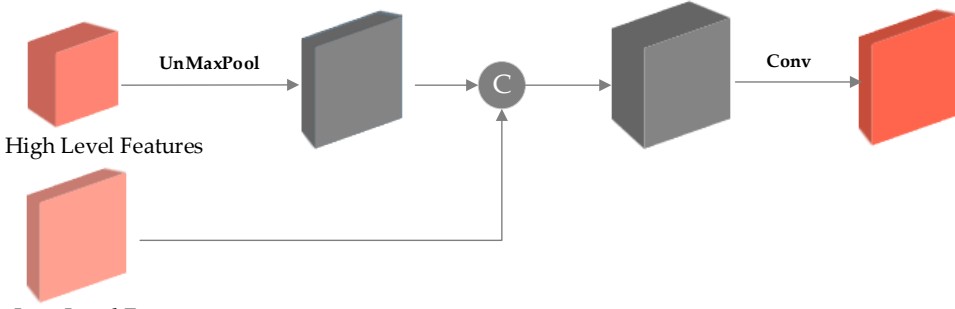

**Figure 5.** Structure of the upsampling fusion module in the decoder stage.

Upon the completion of the upsampling process, the segmentation network seamlessly integrates the feature maps obtained from the three branches, effectively combining their respective information. This fusion process culminates in the generation of a unified feature map, which serves as the final output of the network. By consolidating the information from multiple branches, the segmentation network achieves a comprehensive representation that encapsulates the collective knowledge extracted throughout the upsampling process. The output of the segmentation network can be defined as follows:

$$F_{OUT} = \delta(conv(F_R \oplus F_D \oplus F_F)) \tag{8}$$

### 3.3. Multidimensional Interaction Channel Attention Based 6DoF Object Pose Estimation

In previous research on 6DoF pose detection networks based on CNN feature extraction, leveraging the mask information provided by semantic segmentation networks to extract spatial features of the target objects has been particularly crucial. Inspired by the CBAM [64] algorithm, we began to focus on how to extract spatial feature information of the target objects more accurately from the perspective of attention mechanisms, forming robust feature descriptions for the bounding box regions of the target objects. Therefore, we made improvements to the feature extraction stage of the pose detection network, endowing the network with the capability to adaptively select the number of point cloud samples and incorporating a multi-channel interactive attention module in the feature extraction network to enhance the robust feature representation of the target objects and improve the reliability of pose regression. Figure 6 illustrates the predicting head structure of the 6DoF pose estimation network in this work.

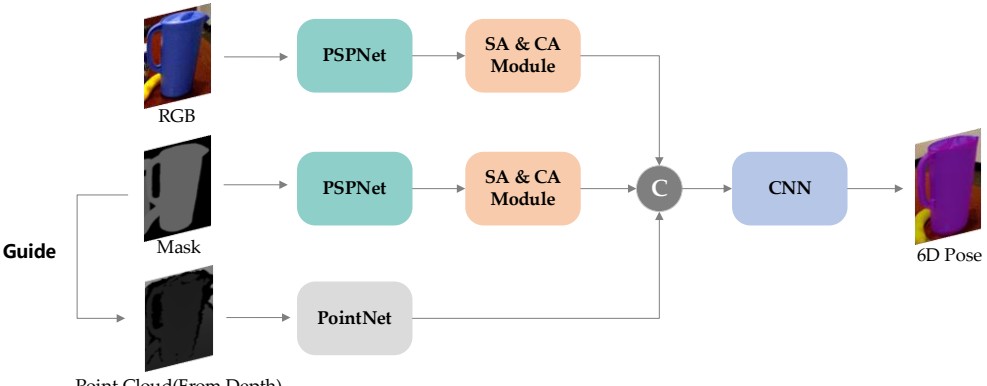

**Figure 6.** Structure of the predicting head in the 6DoF Pose Estimation Network. After obtaining the bounding box for each target object, we used PSPNet to extract dense features from RGB images and Mask images, and PointNet for dense features from point clouds. These features are fused and passed through a four-layer CNN network to regress the 6DoF poses.

### 3.3.1. Feature Extraction Stage

After obtaining the mask image output from the instance segmentation network, we leveraged the semantic information provided by the mask to precisely locate the position of each target object within the image. By utilizing this positional information, we individually cropped the RGB and depth data for each target object and input them into separate neural networks for feature extraction across different data modalities. This approach allowed us to capture and analyze the unique characteristics of each object in a more comprehensive manner.

For the depth feature extraction network, we enhanced the PointNet [39] architecture by incorporating adaptive selection of the number of point cloud samples. Specifically, leveraging the mask image generated by the segmentation network, we determined the bounding box area for each target object, which provides an estimation of the object's physical volume. To accurately represent the spatial pose of small objects, we proposed a method that dynamically increases the number of point cloud samples for objects with smaller bounding box areas, thereby capturing more detailed spatial information. Conversely, for larger objects, the proposed architecture utilizes a normal number of point cloud samples. This adaptive sampling strategy enabled us to effectively capture the intricate spatial characteristics of objects of varying sizes.

For the RGB feature extraction network, we employed a modified version of PSP-Net [65] as the backbone, based on ResNet18 [66], with the last fully connected layer removed. Additionally, we utilized a similar PSPNet approach for processing the binary mask images generated by the segmentation network. However, in this case, the network input consisted of a single channel instead of three. After obtaining these three sets of features, we further enhanced their robustness by feeding them into the CBAM (Convolutional Block Attention Module) module. This module employs both spatial-wise and channel-wise attention mechanisms to augment the feature representation, thereby enhancing the overall precision of the networks's output. Each of the three feature extraction networks is capable of extracting shape features for up to 500 objects, which serve as valuable input for subsequent stages, significantly enhancing the accuracy and effectiveness of our research methodology.

### 3.3.2. 6DoF Pose Estimation and Refinement Stage

In the 6DoF pose estimation stage, we consolidated all the extracted features from each data source into a unified vector, which was then processed through convolutional layers to integrate the diverse feature representations. Subsequently, the proposed method employed an additional neural network to receive the concatenated features and perform regression analysis to estimate the 6DoF pose of the object, encompassing both its rotation matrix and translation vector. Specifically, we concatenated all the extracted features and input them into two distinct neural networks, each comprising four convolutional layers. One network was dedicated to regressing the translation vector, while the other network focused on estimating the rotation matrix.

As shown in Figure 7, the proposed architecture incorporated a pose refinement network, akin to the one employed in DenseFusion. This refinement network takes the output of the 6D pose estimation network as input and enhances the accuracy of the predicted pose. However, it is worth noting that during the testing phase, the benefits of this refinement network become apparent only after it has been trained for approximately 30 epochs. The training process of the refinement network is very time-consuming and requires a lot of computational resources. Nevertheless, once the training is completed, the prediction process becomes relatively faster, enabling efficient and rapid pose estimation.

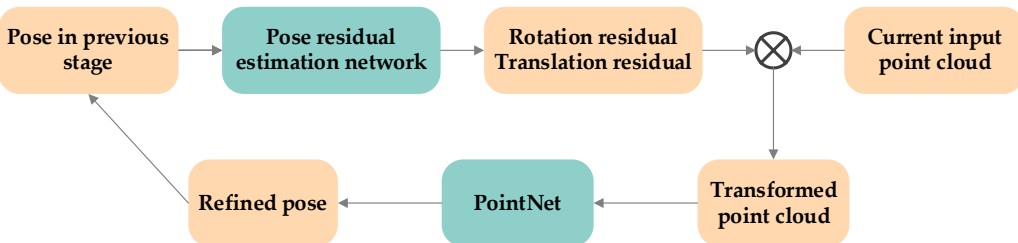

- **Orange blocks indicate data**
- **Green block indicates neutral network**

**Figure 7.** The pose refinement network structure proposed in this study is similar to Densefusion's approach, in which the attitude results of the previous stage are fused with the features of the point cloud data of the current stage, and then the neural network is used for attitude estimation and iteration.

*3.4. 6DoF Object Pose Estimation Network Loss Design*

In order to train our target position detection network, we employed the identical loss function utilized in the Densefusion framework. This loss function plays a pivotal role in guiding the network during the training process, facilitating the accurate detection and localization of target positions. By leveraging this loss function, we ensure that our network learns to effectively align the predicted target positions with the ground truth values, thereby enhancing the overall performance and accuracy of our detection system. The loss is defined in Equation (9).

$$\mathcal{L}_i^p = \frac{1}{M}\sum_j \left\| (Rx_j + t) - (\hat{R}_i x_j + \hat{t}_i) \right\| \tag{9}$$

In Equation (9), $x_j$ represents the number of randomly sampled 3D points from the object model, $M$ and $j$ are the ground truth rotation matrix and translation vector provided by the YCB-Video dataset [67], and $[R_i|T_i]$ are the predicted rotation matrix and translation vector generated by the dense similar fusion embedding of the $i$th input point. $R_i$ represents the predicted rotation matrix, and $T_i$ represents the predicted translation vector. Using this loss function, the pose estimation network can accurately and quickly regress the correct pose of the target.

## 4. Experiments and Results

*4.1. Dataset*

This study evaluated the proposed method on the YCB-Video dataset. The YCB-Video dataset is a large open dataset for 6DoF pose estimation, created in collaboration between the Robotics and State Estimation Lab (RSE-Lab) at the University of Washington and NVIDIA. The dataset consisted of 21 objects, and for each scene, 3 to 9 objects were selected to create a realistic indoor environment. The scenes were captured using RGB-D cameras, resulting in 92 videos, containing a total of 133,827 frames of 640 × 480 images. The dataset was manually annotated with 6DoF poses using a semi-automatic method.

Each frame in the dataset contained up to nine different objects and at least three different objects, with an average of 4.47 objects per frame. The dataset included challenging scenarios with occlusions among objects and low-quality depth maps, making it an ideal benchmark for evaluation.

Furthermore, approximately 60% of the images in the dataset were synthetic, which can improve the model's generalization ability. However, to prioritize training speed, the experiments in this study did not use these synthetic images during the training process. Only real scene images were used.

To facilitate the evaluation using the YCB-Video dataset, the authors of PoseCNN provided the YCB-Video dataset Toolbox in MATLAB. This toolbox enables convenient computation of evaluation metrics like the percentage of ADD-S [49] smaller than 2 cm.

*4.2. Experimental Details*

To fully utilize the label information provided by the YCB-Video dataset, the proposed network in this study was not an end-to-end network. Instead, the instance segmentation network and the pose detection network needed to be trained separately.

During the experimental process, we trained the network using 92 videos and tested it on some keyframes that were not included in the training set.

The experiments were conducted on the PyTorch 1.8 platform using an NVIDIA GeForce RTX3090Ti (24 GB memory). In the training process of the segmentation network, we used the Adam optimizer with an initial learning rate of 0.0001, a batch size of 2, and we trained it for 300 epochs. For the pose detection network training process, we also used the Adam optimizer with an initial learning rate of 0.0001, a batch size of 8, and trained it for 300 epochs.

During the training of the instance segmentation network, we used the training list (16,189 pics) and the testing list (2949 pics) provided by the YCB-Video dataset. In each epoch, we randomly selected 5000 images for training and 1000 images for testing. For training the pose prediction network, we used all the images from both the training and testing lists.

Regarding the conversion of clipped depth images to point cloud data, this study applied adaptive point cloud selection based on the area occupied by the object's bounding box in the image. If the area value was greater than 2000, indicating that the object's projection in 3D space was large enough, we considered using 1000 sparse point cloud points to represent the object's spatial information. Conversely, if the area value was smaller, indicating a smaller volume of the object's projection in 3D space, we used 2000 3D points to fully represent the spatial information of small-volume objects. This strategy is detailed in Table 1.

**Table 1.** Relationship between bounding box area (number of pixels) and number of filtered point clouds.

| Bounding Box Area (Number of Pixels) | Number of Filtered Point Cloud |
|:---:|:---:|
| $\geq$2000 | 1000 |
| <2000 | 2000 |

*4.3. Evaluation Metrics*

We measured the framework's performance using two commonly used 6DoF evaluation metrics: Average Distance of Surface Points (ADD-S) and Percentage of ADD-S below 2 cm (<2 cm). Generally, higher values of these two metrics indicate better framework detection performance.

ADD-S is a pose error measure that is invariant to object symmetry and can comprehensively evaluate both symmetric and asymmetric objects. For a given estimated pose $[R|T]$ and the ground truth pose $[R|T]$, ADD-S computes the average distance of each 3D model point transformed by $[R|T]$ to its nearest neighbor on the target model transformed by $[R|T]$. We report the Area Under the Curve (AUC) of the ADD-S curve, which is the same as in PoseCNN [49].

$$DD - S = \frac{1}{m} \sum_{x_1 \in \ model} min_{x_2 \in \ model} \left\| (Rx_1 + T) - \left( R_p x_2 + T_p \right) \right\| \tag{10}$$

In the formula, *model* represents the sum of 3D model point distances, and *m* is the number of points. $[R|T]$ denotes the ground truth, and $[R_p|T_p]$ represents the predicted values. In this case, we can plot the precision-recall curve and calculate AUC for pose evaluation.

The second metric is the Percentage of ADD-S below 2 cm (<2 cm). Most robot gripper tolerances are within 2 cm, and this metric measures the prediction accuracy under the

minimum robot operation tolerance. We report this percentage to illustrate the performance of the robot in practical operations.

### 4.4. Experiments of the Proposed Segmentation Network

4.4.1. Results of CM and CL Module-Based Segmentation Network

In this study, the proposed segmentation method was individually tested using the YCB-Video dataset as input.

To evaluate the performance of the proposed segmentation algorithm compared to DenseFusion, we utilized five different methods to measure the similarity between the output label image of the instance segmentation network and the ground truth label image. These methods consisted of Mean Hash, Difference Hash, Perceptual Hash, Single-Channel Histogram, and Three-Channel Histogram. It should be noted that smaller hash values indicate better performance, while larger histogram values indicate better results. We calculated the average Siamese distance for 2949 images from the testing list in the YCB-Video dataset, and the results are presented in Table 2.

**Table 2.** Comparison results of different semantic segmentation networks.

| Segmentation Network | aHash | dHash | pHash | One-Channel Histogram | Three-Channel Histogram |
|---|---|---|---|---|---|
| DenseFusion | 1.97 | 2.31 | 5.55 | 98.86 | 97.83 |
| Proposed Method | 0.53 | 0.67 | 1.88 | 99.90 | 98.92 |

The results show that in the YCB-Video dataset, the proposed segmentation network algorithm outperformed DenseFusion network in terms of the similarity between the segmentation map and ground truth map when evaluated using hash algorithms, with an improvement of 1.44% to 3.67%. Similarly, when evaluated using histogram algorithms, there was an improvement of 1.04% to 1.09%. This analysis demonstrates the effectiveness of the proposed network in the instance segmentation of multiple objects, as its ability to learn and distinguish features of multiple objects can significantly aid the subsequent pose estimation process.

In order to more intuitively appreciate the effect of the segmentation network proposed in this study, the visualizations were performed (Figure 8).

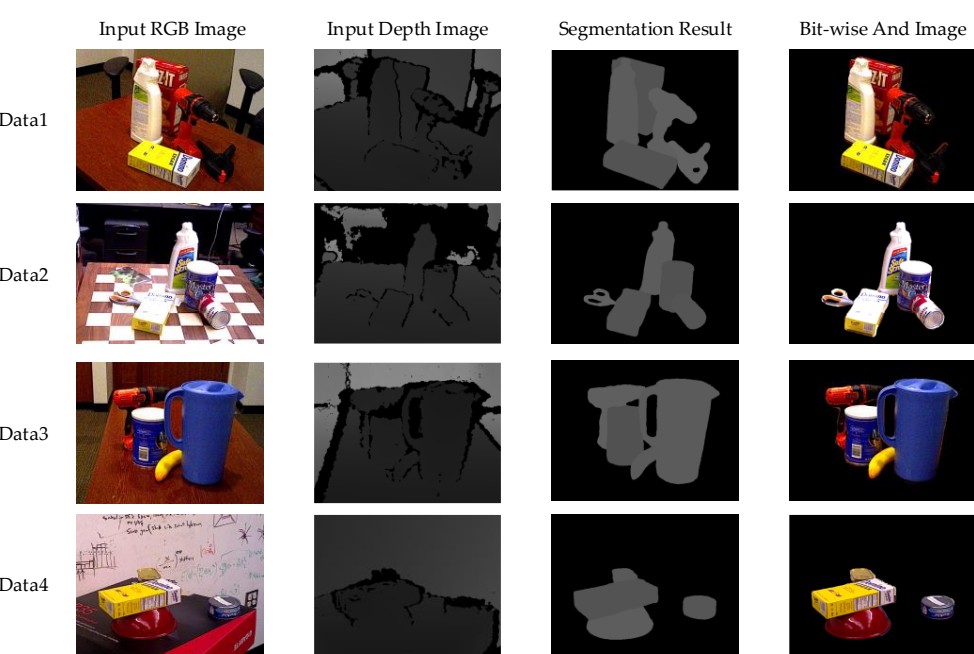

**Figure 8.** Visualization of input and output of the segmentation network.

The proposed segmentation method was highly accurate with sharp edge handling, as evident from Figure 8. Additionally, we conducted speed tests and found that the segmentation can be completed in just 0.06 s on an NVIDIA GeForce RTX3090Ti, meeting the real-time processing requirements of robotic arms.

However, it should be noted that the segmentation network was not entirely perfect in handling all input data. As shown in Figure 9, the proposed network may encounter challenges when dealing with complex edge information, leading to occasional small area misidentifications in complex scenes. Furthermore, since the subsequent pose estimation task heavily relied on segmentation results, inaccuracies in segmentation could potentially impact the accuracy of pose estimation.

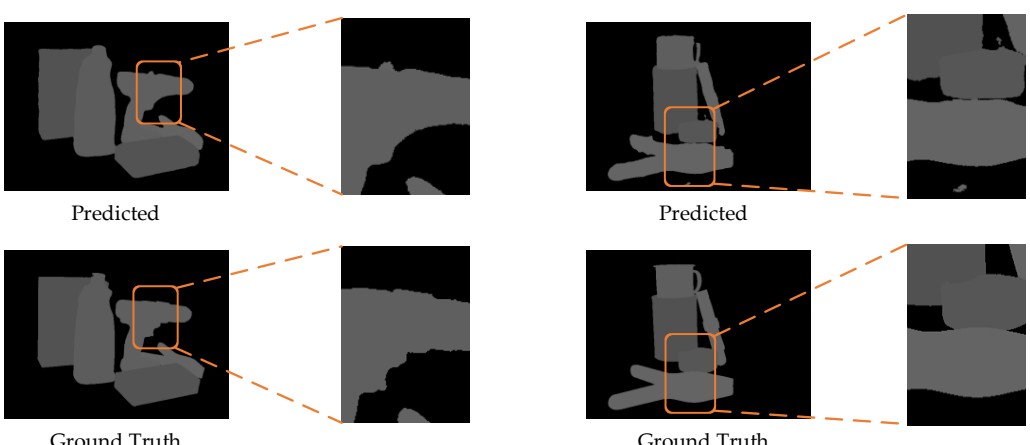

**Figure 9.** Samples with unreliable segmentation results.

### 4.4.2. Ablation Study

In this section, two ablation experiments based on the YCB-Video dataset are listed to evaluate the effectiveness of the instance segmentation algorithm proposed in this study. We examined, in detail, the individual module contributions in the proposed instance segmentation network algorithm. For each part of the ablation experiments, we rigorously retrained the entire network with the same parameter settings.

To evaluate the performance of the CL module and CM module in the proposed segmentation algorithm, we compared the proposed segmentation algorithm with the segmentation algorithm where the CL module and CM module were removed, using the same training parameters. The results are presented in Table 3.

**Table 3.** Comparison of segmentation maps with ground truth labels for the segmentation network proposed in this study and the segmentation networks without the CL or CM modules.

| Segmentation Network | aHash | dHash | pHash | One-Channel Histogram | Three-Channel Histogram |
|---|---|---|---|---|---|
| Proposed Method | 0.53 | 0.67 | 1.88 | 99.90 | 98.92 |
| Proposed Method without CL Module | 0.95 | 1.19 | 3.06 | 99.42 | 98.66 |
| Proposed Method without CM Module | 8.38 | 6.47 | 19.64 | 97.23 | 95.93 |

Table 3 illustrates that the proposed method incorporating CL and CM modules achieves superior performance compared to the other two architectures. These results clearly demonstrate the effectiveness of CL and CM modules in enhancing the accuracy of semantic segmentation.

*4.5. Results of Improved Pose Predict Network*

We evaluated our method on the YCB-Video dataset and compared it with several other methods, including PoseCNN [49]+ICP, PointFusion [68], the baseline DenseFusion [6], the MaskedFusion [7], and the recent FFB6D [5] method.

To assess the performance of the networks, we used the AUC as the evaluation metric, and the results are presented in Table 4. We conducted three repeated experiments, and for the evaluation metric ADD-S AUC, except for PointFusion which performed poorly, all other methods achieved an average score of over 90%. Among the CNN-based networks, our proposed method showed superior performance, especially for objects 051_large_clamp and 052_extra_large_clamp, where our method exhibited an average improvement of 3.6% to 6.4% compared to other methods. This demonstrates that our method has strong discriminative capabilities for object size and excellent feature extraction ability for objects with weak textures. Overall, among the CNN-based feature extraction networks, our proposed method performed the best. Although the average AUC score was not as high as FFB6D, our method outperformed FFB6D for most objects. Only for objects 051_large_clamp and 052_extra_large_clamp, there was a significant performance gap compared to the FFB6D method, which should be a focus of future research.

**Table 4.** Evaluation of 6D Pose (AUC) on the YCB-Video dataset.

| Objects | PointFuion AUC | PoseCNN+ICP AUC | DenseFusion AUC | MaskedFusion AUC | FFB6D AUC | Proposed Method AUC |
|---|---|---|---|---|---|---|
| 002_master_chef_can | 90.9 | 95.8 | 96.4 | 95.5 | 96.3 | 97.5 |
| 003_checker_box | 80.5 | 92.7 | 95.5 | 96.7 | 96.3 | 97.4 |
| 004_sugar_box | 90.4 | 98.2 | 97.5 | 98.1 | 97.6 | 98.0 |
| 005_tomato_soup_can | 91.9 | 94.5 | 94.6 | 94.3 | 95.6 | 94.8 |
| 006_mustard_bottle | 88.5 | 98.6 | 97.2 | 98.0 | 97.8 | 97.5 |
| 007_tuna_fish_can | 93.8 | 97.1 | 96.6 | 96.9 | 96.8 | 98.2 |
| 008_pudding_box | 87.5 | 97.9 | 96.5 | 97.3 | 97.1 | 98.5 |
| 009_geltain_box | 95.0 | 98.8 | 98.1 | 98.3 | 98.1 | 99.0 |
| 010_potted_meat_can | 86.4 | 92.7 | 91.3 | 89.6 | 94.7 | 95.5 |
| 011_banana | 84.7 | 97.1 | 96.6 | 97.6 | 97.2 | 98.6 |
| 019_pitcher_base | 85.5 | 97.8 | 97.1 | 97.7 | 97.6 | 96.8 |
| 021_bleach_cleanser | 81.0 | 96.9 | 95.8 | 95.4 | 96.8 | 95.5 |
| 024_bowl | 75.7 | 81.0 | 88.2 | 89.6 | 96.3 | 88.5 |
| 025_mug | 94.2 | 95.0 | 97.1 | 97.1 | 97.3 | 98.0 |
| 035_power_drill | 71.5 | 98.2 | 96.0 | 96.7 | 97.2 | 97.8 |
| 036_wood_block | 68.1 | 87.6 | 89.7 | 91.8 | 92.6 | 94.6 |
| 037_scissors | 76.7 | 91.7 | 95.2 | 92.7 | 97.7 | 98.4 |
| 040_large_marker | 87.9 | 97.2 | 97.5 | 97.5 | 96.6 | 98.7 |
| 051_large_clamp | 65.9 | 75.2 | 72.9 | 71.9 | 96.8 | 75.0 |
| 052_extra_large_clamp | 60.4 | 64.4 | 69.8 | 71.4 | 96.0 | 72.7 |
| 061_foam_brick | 91.8 | 97.2 | 92.5 | 94.3 | 97.3 | 97.7 |
| MEAN | 83.9 | 93.0 | 93.1 | 93.3 | 96.6 | 94.5 |

Table 5 presents the percentage of ADD-S <2 cm, as the minimum tolerance for robot operation is 2 cm.

Like ADD-S AUC, we repeated the experiments three times to calculate the <2 cm metric. Except for PointFusion, all methods performed well in the <2 cm metric. Our proposed method showed significant improvements for objects 051_large_clamp and 052_extra_large_clamp, once again demonstrating its excellent feature extraction ability for objects with weak textures and its sensitivity to objects of similar sizes. As FFB6D method did not provide the <2 cm metric, it is not reflected in the table.

Figure 10 shows the visualization of our proposed framework for 6DoF pose estimation. It can be observed that the proposed framework achieved relatively high accurate segmentation and pose estimation results for target objects in most scenarios. Although there were some errors, occasional inaccuracies in pose estimation do not matter in real

robot manipulation scenarios because the network can correct them in time in subsequent frames.

**Table 5.** Evaluation of 6D Pose (percentage of ADD-S smaller than 2 cm) on the YCB-Video dataset.

| Objects | PointFuion <br> <2 cm | PoseCNN+ICP <br> <2 cm | DenseFusion <br> <2 cm | MaskedFusion <br> <2 cm | Proposed Method <br> <2 cm |
|---|---|---|---|---|---|
| 002_master_chef_can | 99.8 | 100.0 | 100.0 | 100.0 | 100.0 |
| 003_checker_box | 62.6 | 91.6 | 99.5 | 99.8 | 100.0 |
| 004_sugar_box | 95.4 | 100.0 | 100.0 | 100.0 | 100.0 |
| 005_tomato_soup_can | 96.9 | 96.6 | 96.9 | 96.9 | 96.9 |
| 006_mustard_bottle | 84.0 | 100.0 | 100.0 | 100.0 | 100.0 |
| 007_tuna_fish_can | 99.8 | 100.0 | 100.0 | 99.7 | 100.0 |
| 008_pudding_box | 96.7 | 100.0 | 100.0 | 100.0 | 100.0 |
| 009_geltain_box | 100.0 | 100.0 | 100.0 | 100.0 | 100.0 |
| 010_potted_meat_can | 88.5 | 93.6 | 93.1 | 94.2 | 97.3 |
| 011_banana | 70.5 | 99.7 | 100.0 | 100.0 | 100.0 |
| 019_pitcher_base | 79.8 | 100.0 | 100.0 | 100.0 | 100.0 |
| 021_bleach_cleanser | 65.0 | 99.4 | 100.0 | 99.4 | 99.8 |
| 024_bowl | 24.1 | 54.9 | 98.8 | 95.4 | 100.0 |
| 025_mug | 99.8 | 99.8 | 100.0 | 100.0 | 99.8 |
| 035_power_drill | 22.8 | 99.6 | 98.7 | 99.5 | 99.6 |
| 036_wood_block | 18.2 | 80.2 | 94.6 | 100.0 | 98.8 |
| 037_scissors | 35.9 | 95.6 | 100.0 | 99.9 | 100.0 |
| 040_large_marker | 80.4 | 99.7 | 100.0 | 99.9 | 100.0 |
| 051_large_clamp | 50.0 | 74.9 | 79.2 | 78.7 | 80.9 |
| 052_extra_large_clamp | 20.1 | 48.8 | 76.3 | 75.9 | 82.1 |
| 061_foam_brick | 100.0 | 100.0 | 100.0 | 100.0 | 100.0 |
| MEAN | 74.1 | 93.2 | 96.8 | 97.1 | 97.6 |

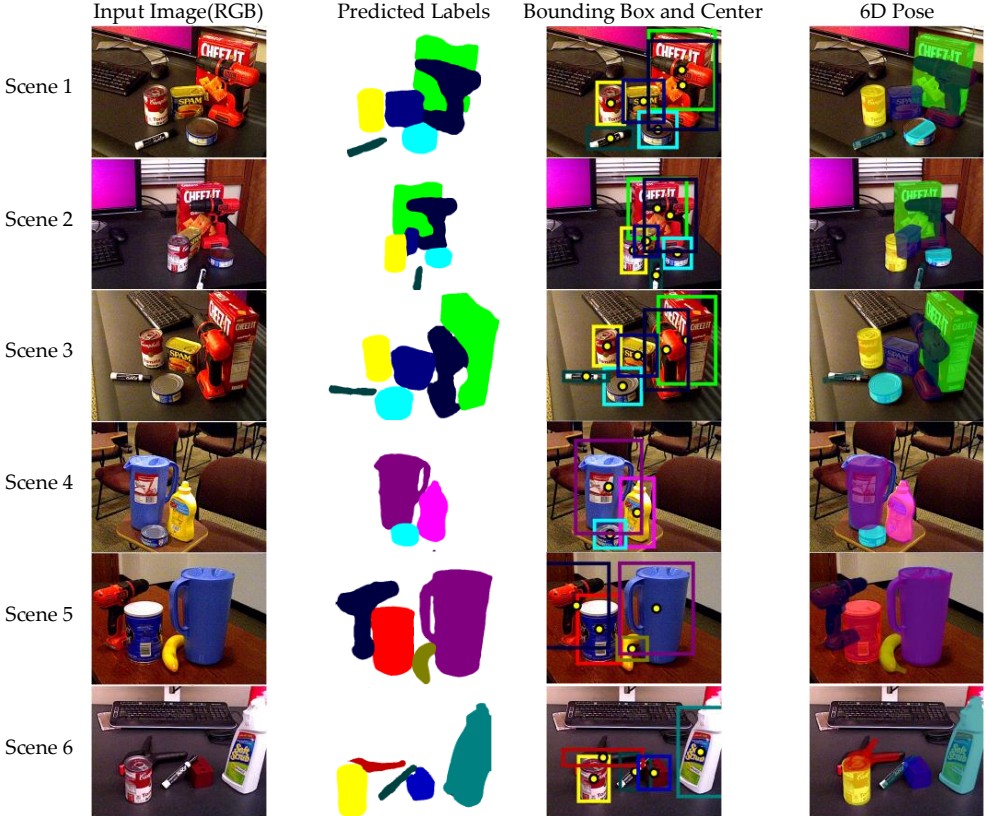

**Figure 10.** Visualization of the overall effectiveness of the framework.

However, due to the segmentation results not always being accurate, the pose estimation also occasionally encountered issues. In Figure 11, we showcase some instances where inaccurate segmentation led to inaccurate pose estimation. Improving segmentation accuracy remains a key focus for future research.

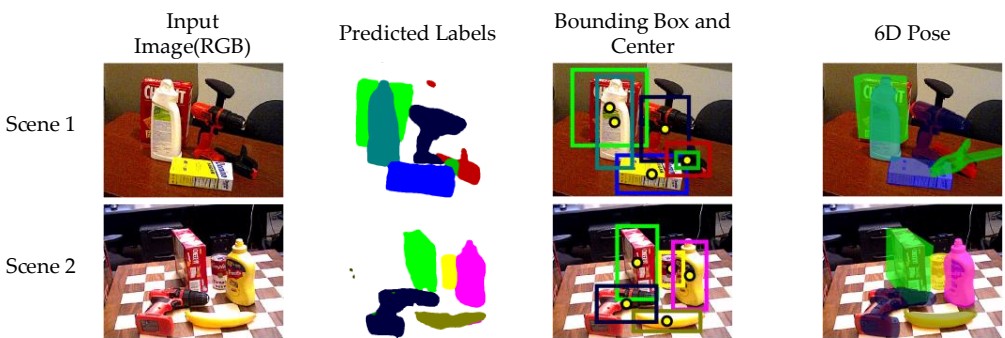

**Figure 11.** Inaccurate segmentation results lead to failed pose detection.

*4.6. Results in Inference Time*

The inference time of the proposed segmentation network in this study was approximately 0.08 s per frame, while the pose prediction network and refinement network had an inference time of approximately 0.012 s per frame. Therefore, the total inference time for obtaining 6DoF pose results from RGB-D capture using our proposed method was approximately 0.092 s per frame.

As shown in Table 6, our two-stage approach achieved better performance and was two times faster compared to the PVN3D method. Moreover, when compared to the MaskedFusion method, which also utilized a two-stage design, our approach exhibited a 2.3 times speed improvement due to the faster segmentation network in the forward process. Through comprehensive testing, the proposed method in this study met the real-time requirements for static object pose estimation in robotic arms.

**Table 6.** Model run-time on the YCB-Video dataset. SS: Semantic Segmentation; PE: Pose Estimation.

| Network | SS (ms/Frame) | PE (ms/Frame) | All (ms/Frame) |
|---|---|---|---|
| PVN3D | 170 | 20 | 190 |
| MaskedFusion | 200 | 12 | 212 |
| Proposed Method | 80 | 12 | 92 |

**5. Conclusions**

In this paper, we addressed the data fusion problem that affects the position estimation of visual targets. Our analysis demonstrated the critical importance of effectively utilizing the information from both the RGB image modality and the depth image modality to achieve an accurate six-degree-of-freedom position estimation of visual targets.

To tackle this challenge, we proposed a two-stage position estimation method based on convolutional neural networks. Our method incorporated a cross-modal cross-level composite feature strategy, which adaptively preserved essential information while reducing the impact of invalid information on target position estimation.

To validate the effectiveness of our method, we conducted a series of comparison experiments with other existing methods, as well as ablation experiments. The experimental results clearly demonstrate that our proposed cross-modal fusion method achieved high accuracy in visual target position detection. Based on the YCB-Video dataset, our approach achieved state-of-the-art performance (94.5%) in terms of average accuracy for 21 objects using a CNN-based methodology. Compared to the classic DenseFusion, our approach demonstrated an average improvement of 1.4%. While our method's average

accuracy for the 21 objects did not surpass the feature-designed FFB6D algorithm due to the limitations of CNN feature extraction capabilities, it outperformed FFB6D for 14 out of the 21 objects, reaching a new state-of-the-art level. Additionally, our proposed method achieved the highest accuracy for the <2 cm metric, indicating its precision in real-world robotic operations.

This study expanded the theoretical groundwork of robot grasping and pose estimation. In the context of instance segmentation, we introduced a novel RGB-D-based segmentation method. For pose estimation, we proposed a new 6DoF pose estimation framework and extensively explored the dense feature fusion phase, incorporating attention mechanisms that significantly enhance performance. This research holds implications across various domains, such as medical robotic assistance, dynamic obstacle avoidance in autonomous driving, and virtual/augmented reality applications.

Although our proposed approach demonstrated exceptional accuracy in 6DoF pose estimation, there is still room for improvement when dealing with objects exhibiting high apparent consistency. We aim to further investigate such objects and refine our modeling strategies. Moreover, we recognize the need for advancements in the technique of assigning 3D rotations and translations to each sampled point, followed by the derivation of final pose data based on confidence levels.

**Author Contributions:** Conceptualization, Z.W. and X.S.; methodology, Z.W. and H.W.; software, H.W., Q.M. and Q.Z.; writing—original draft preparation, Z.W. and Q.Z.; writing—review and editing, H.W. and X.S.; visualization, Q.M. and Q.Z. All authors have read and agreed to the published version of the manuscript.

**Funding:** This research was funded by the National Natural Science Foundation of China (grant number 61903162) and Jiangsu Province Industry University Research Cooperation Project: Research on key technology of robot navigation and following in human–robot integration environment (BY20221143).

**Data Availability Statement:** The data presented in this study are available on request from the corresponding author.

**Conflicts of Interest:** The authors declare no conflict of interest.

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
