# Peer review of "Enhancing 6-DoF Object Pose Estimation through Multiple Modality Fusion: A Hybrid CNN Architecture with Cross-Layer and Cross-Modal Integration"

_machines, doi:10.3390/machines11090891_

Round 1

Reviewer 1 Report

The paper is well written, clear and relevant to the community.

I would like the authors to expand more on the interpretability part and, for instance, on the use of post-hoc interpretability to understand how the model is using data. I believe that this is an important and virtually unexplored area.

In fact, it may be useful that the authors take a look at the following references and add a paragraph of discussion on this topic and references:

- https://www.nature.com/articles/s42256-023-00620-w 

- https://dl.acm.org/doi/pdf/10.1145/3236386.3241340

English is satisfactory

Author Response

We gratefully thanks for the precious time the reviewer spent making constructive remarks. We have carefully considered all comments and revised our manuscript accordingly. The detailed response are as follows:

Point 1: The paper is well written, clear and relevant to the community.

Response 1: We thank the reviewer for reading our paper carefully and giving the above positive comments.

Point 2: I would like the authors to expand more on the interpretability part and, for instance, on the use of post-hoc interpretability to understand how the model is using data. I believe that this is an important and virtually unexplored area.

Response 2: We greatly appreciate the constructive feedback from the reviewer. We believe that post hoc interpretability is essential for enhancing people's understanding of deep learning models. Moreover, we find alignment between this perspective and the underlying concept of our ablation experiments. As a result, we have made revisions in Page1, as indicated in line 91-94, to address this aspect.

Point 3: In fact, it may be useful that the authors take a look at the following references and add a paragraph of discussion on this topic and references:

- https://www.nature.com/articles/s42256-023-00620-w 

- https://dl.acm.org/doi/pdf/10.1145/3236386.3241340

Response 3: We extend our sincere gratitude to the reviewer for highlighting the post hoc interpretability approach. We have carefully studied the two articles you have recommended and recognized their significant relevance to our work. In acknowledgment of this, we have incorporated citations to these two articles in Page 2, as indicated in line 93.

Reviewer 2 Report

The article discusses the problem of combining goal visualization data. The article also provides a comparison of this method with other existing methods. This is a good article, but there are a few comments:

1. It is necessary to elaborate on the proposed method in more detail.

2. In conclusion, you need to describe the result, not what has been done.

3. What is the scientific significance?

4. In which fields of science can this technique be applied.

Minor editing of English language required

Author Response

Thank you very much for giving us the opportunity to revise our manuscript, we appreciate the careful review and valuable suggestions. We have carefully considered all comments and revised our manuscript accordingly. The detailed response are as follows:

Point 1: The article discusses the problem of combining goal visualization data. The article also provides a comparison of this method with other existing methods. This is a good article.

Response 1: We are truly grateful for the reviewer's thorough reading of our paper and the recognition of the methodology presented in our article.

Point 2: It is necessary to elaborate on the proposed method in more detail.

Response 2: We deeply appreciate your thorough review of our proposed method. Upon further review of our article, we acknowledge that there were certain areas where the description of our methodology lacked detail. To address this, we have now provided references to specific sections for each component of the network framework (refer to Page 5, line 236-239). Additionally, we have elaborated on the U-Net architecture (refer to Page 6, line 248-250). Furthermore, we have updated the description of the Pose Refine Network and included a schematic of the fine-tuning network (Figure 7). We have also provided a comprehensive explanation of the role of refinement (refer to Page 11, line 454-465).

Point 3: In conclusion, you need to describe the result, not what has been done.

Response 3: We genuinely appreciate your attention to our experimental results. In terms of our experimental findings, we have readdressed them in Chapter 5. By comparing the average accuracy and <2cm precision across the 21 objects in the YCB Video dataset, as well as comparing with other methods, we substantiate the effectiveness of our proposed approach. For a more detailed insight, please refer to lines 669 to 677 on page 18.

Point 4: What is the scientific significance?

Response 4: We sincerely appreciate your feedback regarding our conclusion section. In terms of the scientific significance of the method proposed in this paper, we firmly believe that it extends the boundaries of research in both image segmentation and robot pose estimation domains. For more details, please refer to Page 19, line 678-684.

Point 5: In which fields of science can this technique be applied.

Response 5: We genuinely appreciate your concern for the potential future extensions of this research. We firmly believe that the proposed method holds constructive contributions for domains such as medical robotics and autonomous driving. For further elaboration, please refer to Page19, line 682-684.

Reviewer 3 Report

1. Please describe the introduction in more detail. There is no systematicization. A clear indication of similar solutions, their advantages and disadvantages.

2. In Conclusion, please clearly indicate the possibility of developing the method and further plans of the authors.

Moderate editing of English language required.

Author Response

We extend our sincere gratitude for affording us the chance to revise our manuscript. We highly value the meticulous review and the invaluable suggestions you provided. We have thoroughly analyzed all the comments and subsequently made revisions to our manuscript. The specific responses are outlined as follows:

Point 1: Please describe the introduction in more detail. There is no systematicization. A clear indication of similar solutions, their advantages and disadvantages.

Response 1: We greatly appreciate your thorough reading of my introduction section. We have made revisions to the introduction, aligning it more closely with the requirements for systematic presentation. The specific modifications can be found between line 39 and line 94 on pages 1-2. Furthermore, regarding the advantages and disadvantages of various solutions, we have addressed this aspect in Chapter 2. Specific instances can be found, for example, between lines 123 and 127 in Chapter 3.

Point 2: In Conclusion, please clearly indicate the possibility of developing the method and further plans of the authors.

Response 2: We truly appreciate your consideration for our future plans. We believe that the method proposed in this study holds extensive applications in areas such as autonomous driving and virtual reality, as elaborated in lines 682 to 684 on page 19.

As for our future research directions, we acknowledge the potential benefits of reevaluating our modeling approach for objects with higher apparent consistency. Moreover, we believe that exploring alternative methods for pose regression is worth investigating, as discussed in lines 685 to 690 on page 19.

Round 2

Reviewer 3 Report

Thanks for the answers.

Moderate editing of English language required.